

# Effectiveness of *Rhizophagus intraradices* and *Acinetobacter calcoaceticus* on soybean growth and thiram residues in soybean grains and rhizosphere soil

Weiguang Jie[1,2,3], Yiwen Tan[1,2], Houze Lin[3], Min Zhang[1,2] and Lianbao Kan[1,2]

[1] Engineering Research Center of Agricultural Microbiology Technology, Ministry of Education & Key Laboratory of Microbiology, College of Heilongjiang Province & School of Life Sciences, Heilongjiang University, Harbin, China

[2] Heilongjiang Provincial Key Laboratory of Plant Genetic Engineering and Biological Fermentation Engineering for Cold Region & School of Life Sciences, Heilongjiang University, Harbin, China

[3] School of Food Engineering, Heilongjiang East University, Harbin, China

Corresponding authors
Weiguang Jie,
jieweiguang2007@126.com
Lianbao Kan, kanlian-
bao2008@163.com

## ABSTRACT

Soybean root rot has a serious effect on soybean yield. Pesticides such as thiram are used to prevent soybean root rot, but thiram remains in the soil, which seriously threatens food safety and human health. Microbial fertilizers can effectively control root rot, promote crop growth, and degrade pesticide residues. This study aimed to evaluate the synergistic effects of arbuscular mycorrhizal fungi and phosphorus-solubilizing bacteria in a controlled environment, specifically investigating their potential for prevention and control of soybean root rot and pesticide degradation. In this pot-based study, we investigated the effects of *Rhizophagus intraradices*, *Acinetobacter calcoaceticus*, and thiram on the incidence of root rot, soybean biomass, the number of bacterial colonies in rhizosphere soil, and thiram residues in soybean grains and rhizosphere soil. The results showed that inoculation with *R. intraradices* and *A. calcoaceticus* significantly increased arbuscular mycorrhizal fungi spore density (445%), arbuscular mycorrhizal fungi infection rate (103%), soybean biomass such as fresh weights (59%), nodule number (237%), and total bacterial colony number in the rhizosphere soil of soybean plants (133%) and presented the lowest incidence of root rot (20%) ($P < 0.05$), compared with the control group. A single inoculant significantly reduced the residual amount of thiram in soybean grains and rhizosphere soil, and a mixed inoculation (*R. intraradices* and *A. calcoaceticus*) produced the most significant reduction, relative to the group sprayed with only thiram, thiram residues were reduced by 73% and 69%, respectively ($P < 0.05$). These findings provide a foundation for the biological control of soybean root rot and the degradation of pesticides and contribute to the sustainable development of agricultural ecosystems.

## INTRODUCTION

Soybeans are rich in nutrients, including protein, fat, carbohydrates, and a variety of vitamins that are essential to people in their daily lives (*Medic, Atkinson & Hurburgh, 2014*;

*Specht et al., 2015*). In recent years, soybean planting and yields have increased steadily, but soybean root rot can cause crop reduction or even collapse, causing considerable economic losses (*Yu et al., 2022*). Soybean root rot is a soil-borne fungal disease that endangers soybean production. Owing to its severe symptoms, a range of pathogenic fungi are involved, and the difficulty in controlling it, soybean root rot has become the main factor restricting soybean production at present (*Giachero, Declerck & Marquez, 2022*; *Rahman, Rubayet & Bhuiyan, 2020*). Soybean root rot is caused by a variety of pathogens, such as *Fusarium*, *Phytophthora*, *Pythium*, *Rhizoctonia*, and *Sclerotium* (*Giachero, Declerck & Marquez, 2022*; *Rahman, Rubayet & Bhuiyan, 2020*). For many years, soybean root rot has been controlled and prevented by spraying organic pesticides. Thiram is a commonly used organic sulfur low-toxicity fungicide with broad-spectrum antibacterial properties; it is a common pesticide for the prevention and control of soybean root rot. However, toxicological experiments have shown that thiram is cytotoxic and teratogenic (*Chouhan et al., 2023*), and it is not easily degraded in the natural environment, which results in soil and groundwater pollution. Thiram residues in the soil may affect the growth, reproduction, and metabolism of indigenous microorganisms, thereby affecting soil biochemical processes such as biological nitrogen fixation, nitrification and denitrification, decomposition of organic matter, sulfur oxidation, and soil nutrient activation (*Liu et al., 2022*). Microorganisms can use enzymatic reactions or nonenzymatic reactions to degrade pesticide residues in crops and the environment, and microbial fertilizer can promote crop growth, improve crop resistance, reduce the use of pesticides, reduce environmental pollution, and improve crop yield (*Kumawat, Razdan & Saharan, 2022*; *Munir et al., 2022*).

Arbuscular mycorrhizal fungi (AMF) are the most widely distributed fungi in the soil and form symbiotic relationships with 80% of terrestrial plants (*Berdeni et al., 2018*). AMF can improve plant nutrient absorption, resilience, disease resistance, and soil structure and fertility (*Hata, Kobae & Banba, 2010*; *Smith, Smith & Jakobsen, 2003*). *Berdeni et al. (2018)* inoculated AMF into the soil and reported that AMF improved the biomass, nutritional status, and disease resistance of apples and that there was a strong interaction between plants above ground and below ground, which improved the disease resistance of apples to pathogenic microorganisms. Our research group's previous studies have shown that inoculation of AMF significantly decreased the disease index of soybean root rot (*Jie et al., 2022*; *Jie et al., 2023*). AMF not only improve soil and plant health but also degrade pesticide residues in soil and crops, which positively impacts food safety and environmental restoration. For example, *Wang et al. (2011)* explored the effects of AMF on crop yield and pesticide residues in soil and crops and reported that AMF reduced phoxim residues, especially in the edible parts of crops. Approximately 30–80% of phosphorus in the soil is in an organic form and cannot be used by plants (*Dalai, 1977*). The main type of organic phosphorus in the soil is phytic acid (*Turner et al., 2002*). However, AMF lack the gene encoding phytase and cannot transform large amounts of organic nutrient elements in the soil (*Zhang et al., 2022*).

Phosphorus-solubilizing bacteria activate soil insoluble inorganic phosphorus and organic phosphorus by secreting metabolites such as protons, organic acids, cations, and phosphatases that increase soil available phosphorus for plants and AMF, which reduces

the need for phosphorus fertilizer and promotes plant growth (*Richardson & Simpson, 2011*). *Ramanjaneyulu et al. (2021)* applied phosphorus solubilizing bacteria together with phosphate fertilizer and reported that the two had a synergistic effect on crop growth. *Chouhan et al. (2023)* analysed the degradation of formebidium by *Pseudomonas otitis* and *Pseudomonas stutzeri* (which have a phosphorus solubilizing effect) and reported that 81% and 57% of formebidium, respectively, was degraded four days after bacterial infection, and 99% of formebidium could be degraded eight days after bacterial infection. *Zhang et al. (2021)* reported that the PSB strain N3 could abate the toxic effects of 4-chloro-2-methylphenoxy acetic acid to improve tomato seedling growth. Increasing soybean yield, reducing pesticide use, and reducing pesticide residues are urgent problems. However, few studies have focused on using AMF and phosphorus solubilizing bacteria to control root rot in soybean and degrade pesticide residues (*Jie et al., 2023*).

In this study, we examined the effects of *R. intraradices*, *A. calcoaceticus*, and thiram on soybean biomass, the incidence of root rot, the total number of bacteria in rhizosphere soil, and the residual amount of thiram in soybean grains and rhizosphere soil. Our findings provide a means to increase soybean disease resistance and yield, lay a foundation for subsequent research on biological agents, and improve our understanding of pesticide degradation, thus improving soybean grain and oil quality and ensuring food safety.

# MATERIALS AND METHODS

## Soybean variety

Heinong 48 soybean variety (disease-susceptible type, protein content 45.23%, fat content 19.5%), purchased from Heilongjiang Academy of Agricultural Sciences, was used in this study.

## AMF inoculants and phosphorus solubilizing bacteria inoculants

We isolated the AMF (*R. intraradices*) from soybean fields in Harbin, Heilongjiang Province. The identification of *R. intraradices* was determined according to the method of *Jie, Cai & Ge (2012)*. The biological characteristics of *R. intraradices* were similar to those of *Yang et al. (2020)*. Using alfalfa as the host plant, the AMF was propagated by the method of *Jie et al. (2021)*.

We obtained the phosphorus solubilizing bacteria (PSB) (*A. calcoaceticus*) from a soybean field in Harbin, Heilongjiang Province. Approximately one g of soil sample was transferred to a 250 mL flask containing 90 mL of sterile water. The mixture was shaken at 170 rpm in a 28 °C constant-temperature incubator for 20 min, followed by 10 min sedimentation to facilitate particle settling. The resulting soil suspension was subsequently subjected to serial gradient dilution. Aliquots (0.1 mL) from appropriate dilution gradients were aseptically spread onto organic phosphorus Montgomery solid medium (glucose 10 g, ferrous sulfate 0.03 g, manganese sulfate 0.03 g, lecithin 0.2 g, sodium chloride 0.3 g, potassium chloride 0.3 g, ammonium sulfate 0.5 g, calcium carbonate 3.0 g, agar 20 g, distilled water one L, pH 7.2–7.4). All plates were incubated at 28 °C for 3–5 days. Distinct colonies exhibiting characteristic transparent phosphorus-solubilizing halos were carefully selected for further purification using the three-zone streaking method. This sequential

purification process was repeated until pure cultures were obtained. The selected PSB were identified by morphological, physiological and biochemical characteristics and 16S rDNA sequence analysis. The PSB was inoculated in LB liquid medium (28 °C for 24 h). The medium was centrifuged at 5,000 r/min for 5 min, bacterial precipitates were collected, and sterile water was added to prepare a bacterial suspension with a number of bacteria $1 \times 10^8$ cfu/mL.

## Experimental design

Pot cultures were used in the experiment, and the soil was obtained from a soybean field (126°65′E, 45°57′N) in Pingfang District of Harbin, Heilongjiang Province. The following eight treatment groups were established: blank control (non-inoculation with *R. intraradices* and *A. calcoaceticus*, non-spraying thiram) (CK); inoculation with *R. intraradices* (R); inoculation with *A. calcoaceticus* (A); inoculation with *R. intraradices* and *A. calcoaceticus* (RA); thiram spraying only (TK); thiram spraying and inoculation with *R. intraradices* (TR); thiram spraying and inoculation with *A. calcoaceticus* (TA); and thiram spraying and inoculation with *R. intraradices* and *A. calcoaceticus* (TRA).

Potted plants were used in the experiment. Each treatment was set up with 10 replicates, and the experiment was designed with a completely randomized block design. No other fertilizers were applied throughout the experiment. Each pot (12 L volume, top diameter of 30 cm, bottom diameter of 23 cm and height 22 cm) contained approximately 12 kg of soil and six soybean seeds, of which three seedlings were left after germination. The basin was surrounded by soybean as a line of protection, to reduce the marginal effect. To ensure that the experiment had practical importance, the production management mode was the same as that used in the adjacent field. The pots were irrigated daily at growth periods. All the soybean plants were grown under controlled conditions: temperature of 23 ± 1 °C, photoperiod of 12 h and humidity > 60%.

The *R. intraradices* inoculum was added before soybean sowing. After 50 g of *R. intraradices* inoculum was added to each pot, it was covered with 1–2 cm of soil, and an appropriate amount of water was sprayed. Then, the soybeans were sown and covered with 1–2 cm of soil, and an appropriate amount of water was sprayed. The *A. calcoaceticus* inoculum was added on the 7th day after the emergence of the soybean plants, at five mL/plant *via* root irrigation; that is, the inoculum was injected near the rhizosphere soil with a syringe, and the other treatment groups were injected with the same volume of sterile water.

Prepare a thiram suspension by adding three g of thiram wettable powder to one L of water. After thorough mixing, evenly spray the suspension onto the rhizosphere soil at 60 d and 90 d after soybean emergence.

## Sample collection

Three soybean plants and their rhizosphere soil, roots, stems, and seeds were randomly selected 120 days after emergence under the different experimental treatments according to the methods of *Zhou et al., (2011)*.

### Determination of AMF spore density and AMF infection rate in rhizosphere soil of soybean plants

AMF spore density and AMF infection rate were determined at 120 d after soybean emergence as previously described in *Jie et al., (2021)*.

### Determination of root rot incidence in soybean plants

The incidence of soybean root rot was recorded by observing the symptoms of the soybean plants in the field; the disease was evident in the soybean hypocotyl and root and in the necrosis of the stem and cotyledon. The soybean root rot disease index was determined as previously described in *Zhou et al. (2011)*.

### Determination of nodule numbers in soybean plants

The number of nodules was counted as previously described in *Zhou et al. (2011)*.

### Analysis of soybean plant biomass

The height, stem diameter, root length, fresh weight, aboveground dry weight, underground dry weight, and yield of each soybean plant was measured as previously described in *Jie et al. (2021)*.

### Determination of total soil bacterial colonies

A total of 25 g of wet rhizosphere soil was weighed, 225 mL of sterile saline solution was added, and the solution was mixed at 180 r/min for 30 min, after which the sample was repeatedly homogenized and diluted 10 times. The specific method can be referred to the previously description of *Jie et al. (2023)*.

### Determination of thiram residue in soybean grains and rhizosphere soil

Thiram standard (50 mg) was weighed and placed in a 100 mL volumetric bottle, to which five mL of methanol and one mL of methanol acetic acid were added. After the soybean grains were crushed with a grinder and the soil was sifted (40 mesh), the soybean grain samples and soil samples were weighed to 10 g (accurate to 0.0001 g), and 50 mL of methanol and acetic acid were added to the mixed solvent (methanol:acetic acid = 9.5:0.5) and then shaken on an oscillator for 1 h. After collection with methanol washing filter paper, the mixture was centrifuged at a speed of 2,000 r/min, and 3.0 mL of the supernatant was absorbed. The mixture was transferred to a 10 mL bottle and methanol was added to bring the volume to 10 mL. The mixture was shaken well and placed in an ultrasonic cleaning machine to fully dissolve the active ingredients. An organic system filter membrane (>0.45 μm) was used to determine the residual amount of thiram. An Agilent HPLC1260 liquid chromatograph was used.

The chromatographic operating conditions were as follows: the mobile phase was methanol:distilled water at a ratio of 65:35 (V:V); the flow velocity was 1.0 mL/min; the detection wavelength was 280 nm; sample quantity was 10 μL; column temperature was 27 °C; and retention time was 4.7 min.

The standard sample and the sample to be tested were loaded into the automatic injector, and after the baseline was stable, each sample was injected three times consecutively as a

technical repetition. The residual amount of Fumai dual in the sample to be tested was calculated by the mass fraction X:

$$X = 100 \, X = 100 \times a_2 m_1 \, p / a_1 m_2$$

where $a_1$ is the thiram peak area in the standard sample; $a_2$ is the thiram peak area in the sample; $m_1$ is the mass of thiram in standard sample (g); $m_2$ is the mass of thiram in the sample (g); and $p$ is the mass fraction of thiram in standard sample solution (%).

## Data processing

The experimental data were analysed using SPSS 27.0 (IBM Corp., Armonk, NY, USA). Analysis of variance (ANOVA) and Duncan's test (honestly significant difference, HSD) were applied to evaluate significant differences ($P < 0.05$). The results are expressed as the means $\pm$ the standard error.

# RESULTS

## Effects of treatments on AMF spore density, AMF infection rate, the incidence of soybean root rot and plant growth parameters

Under no pesticide treatment, AMF spore density in the *R. intraradices* and *A. calcoaceticus* inoculation group was significantly greater than that in the other treatment groups ($P < 0.05$). In contrast to the group inoculated with *R. intraradices*, the group inoculated with *A. calcoaceticus*, and the control group, spore density in the *R. intraradices* and *A. calcoaceticus* inoculation group increased by 70%, 193% and 445%, respectively (Table 1). Under the same inoculation treatment, spraying thiram caused spore density to decrease significantly, indicating that thiram may affect the soil microenvironment, inhibit or kill some of the original soil AMF ($P < 0.05$).

As can be seen from Table 1, the AMF infection rate was the highest in the group inoculated with *R. intraradices* and *A. calcoaceticus*. Compared with the control group, the AMF infection rate of the group inoculated with *R. intraradices*, the group inoculated with *A. calcoaceticus*, and the group inoculated with *R. intraradices* and *A. calcoaceticus* was significantly increased by 91%, 27% and 103% ($P < 0.05$) (Table 1). The results indicated that the synergistic effect of *R. intraradices* and *A. calcoaceticus* was beneficial to increase the AMF infection rate of soybean roots. Under the same inoculation treatment, the AMF infection rate of soybean roots in the thiram spraying group was significantly lower than that in the thiram non-spraying group ($P < 0.05$). The AMF infection rate in the control group was 10% higher than that in the thiram spraying group. The AMF infection rate in the group inoculated with *R. intraradices* and *A. calcoaceticus* was 13% higher than that in the group subject to thiram spraying and inoculated with *R. intraradices* and *A. calcoaceticus*. This may be because thiram causes toxicity to the rhizosphere microorganisms of soybean roots, resulted in the decrease of the AMF infection rate.

Compared with soybean in the other treatment groups, the incidence of root rot disease in the *R. intraradices* and *A. calcoaceticus* treated plants decreased significantly ($P < 0.05$) (Table 1). The blank control group presented the highest incidence of root rot (75%), and the *R. intraradices* and *A. calcoaceticus* inoculation group presented the lowest incidence of

**Table 1** Effects of treatments on the AMF spore density, AMF infection rate incidence of soybean root rot, nodule number and total number of bacterial colonies.

| Treatments | AMF spore density per gram of soil | AMF infection rate (%) | Incidence of soybean root rot (%) | Nodule number per plant | Total number of bacterial colonies (CFU/g) |
|---|---|---|---|---|---|
| R | $2.34 \pm 0.13^b$ | $90.90 \pm 0.85^b$ | $31.67 \pm 0.03^{de}$ | $70.67 \pm 2.52^c$ | $(1.14 \pm 0.80) \times 10^{6b}$ |
| A | $1.36 \pm 0.22^d$ | $60.32 \pm 0.83^e$ | $38.33 \pm 0.03^{cde}$ | $62.00 \pm 2.65^d$ | $(9.05 \pm 0.18) \times 10^{5c}$ |
| RA | $3.98 \pm 0.23^a$ | $96.39 \pm 0.69^a$ | $20.00 \pm 0.00^f$ | $109.00 \pm 2.65^a$ | $(1.66 \pm 0.10) \times 10^{6a}$ |
| CK | $0.73 \pm 0.05^f$ | $47.52 \pm 0.98^g$ | $75.00 \pm 0.05^a$ | $32.33 \pm 4.16^f$ | $(7.12 \pm 0.08) \times 10^{5e}$ |
| TR | $1.08 \pm 0.04^e$ | $82.44 \pm 0.77^d$ | $40.00 \pm 0.05^{cd}$ | $51.33 \pm 2.08^e$ | $(6.65 \pm 0.13) \times 10^{5e}$ |
| TA | $0.43 \pm 0.05^g$ | $52.09 \pm 0.24^f$ | $41.67 \pm 0.03^{bc}$ | $41.00 \pm 4.00^f$ | $(5.87 \pm 0.15) \times 10^{5f}$ |
| TRA | $1.63 \pm 0.11^c$ | $85.33 \pm 0.40^c$ | $30.00 \pm 0.05^e$ | $82.67 \pm 3.51^b$ | $(8.49 \pm 0.13) \times 10^{5d}$ |
| TK | $0.23 \pm 0.04^g$ | $43.39 \pm 0.46^h$ | $50.00 \pm 0.10^b$ | $33.00 \pm 2.00^f$ | $(3.67 \pm 0.06) \times 10^{5g}$ |

**Notes.**

CK, Blank control; R, Inoculation with *R. intraradices*; A, inoculation with *A. calcoaceticus*; RA, inoculation with *R. intraradices* and *A. calcoaceticus*; TK, thiram spraying only; TR, thiram spraying and inoculation with *R. intraradices*; TA, thiram spraying and inoculation with *A. calcoaceticus*; TRA, thiram spraying and inoculation with *R. intraradices* and *A. calcoaceticus*.

Values are means ± standard error with three replicates.

Analysis of variance (ANOVA) and Duncan's test (honestly significant difference, HSD) were applied to evaluate significant differences ($P < 0.05$).

Different lowercase letters indicate significant differences from different treatments ($P < 0.05$).

root rot (20%). The results revealed that *R. intraradices* and *A. calcoaceticus* interacted with each other to effectively inhibit soybean root rot pathogens, thereby reducing the incidence of soybean root rot. Under the same inoculation treatment, spraying thiram caused the incidence of soybean root rot to be significantly greater ($P < 0.05$). The incidence of soybean root rot in the group that was sprayed with thiram and inoculated with *R. intraradices* and *A. calcoaceticus* was 40% lower than that in the thiram spraying only group.

The number of nodules in the *R. intraradices* and *A. calcoaceticus* inoculation group was significantly greater than that in the other treatment groups, compared with the group inoculated with *R. intraradices*, the group inoculated with *A. calcoaceticus*, and the control group, the increase in nodule number was 54%, 76%, and 237%, respectively ($P < 0.05$) (Table 1). The number of nodules in the thiram treatment group was significantly lower than that in the *R. intraradices* and *A. calcoaceticus* inoculation group, indicating that thiram caused stress to the soil microbial environment, inhibited or killed rhizobia in the soil, and reduced the nitrogen fixation ability of soybean plants, resulting in a decrease in the number of nodules ($P < 0.05$).

Plant height, stem diameter, root length, fresh weight, dry weight, and plant yield in the *R. intraradices* and *A. calcoaceticus* inoculation group were significantly greater than those in the single inoculation groups (inoculation with *R. intraradices* or *A. calcoaceticus* alone) ($P < 0.05$) (Table 2). For example, fresh weights in the *R. intraradices* and *A. calcoaceticus* inoculation group increased by 11%, 20%, and 59%, respectively, compared with fresh weights in the *R. intraradices* inoculation group, the *A. calcoaceticus* inoculation group and the control group. Under the same inoculation treatment, plant height, stem diameter, root length, fresh weight, dry weight, and plant yield after spraying thiram were significantly lower than those in the non-spray thiram treatment group ($P < 0.05$).

**Table 2** Effects of treatments on soybean biomass.

| Treatmens | Plant height (cm) | Stem diameter (mm) | Root length (cm) | Fresh weight (g) | Aboveground dry weight (g) | Underground dry weight (g) | Yield per plant (g) |
|---|---|---|---|---|---|---|---|
| R | 60.90 ± 1.67[b] | 6.34 ± 0.20[b] | 25.53 ± 0.66[c] | 65.63 ± 2.08[bc] | 25.40 ± 1.01[c] | 3.47 ± 0.15[bc] | 23.69 ± 0.14[b] |
| A | 58.63 ± 0.73[c] | 6.26 ± 0.16[b] | 23.97 ± 0.47[d] | 60.97 ± 3.44[c] | 24.50 ± 1.49[cd] | 3.17 ± 0.51[c] | 22.00 ± 0.16[c] |
| RA | 66.07 ± 1.60[a] | 6.80 ± 0.07[a] | 30.17 ± 0.81[a] | 72.90 ± 2.49[a] | 31.30 ± 0.91[a] | 4.4 ± 0.26[a] | 26.97 ± 0.26[a] |
| CK | 55.63 ± 0.61[e] | 5.52 ± 0.28[d] | 20.03 ± 0.75[e] | 45.73 ± 3.19[e] | 18.37 ± 1.04[f] | 2.40 ± 0.20[d] | 19.61 ± 0.39[e] |
| TR | 58.07 ± 1.82[cd] | 6.04 ± 0.22[bc] | 23.53 ± 1.06[d] | 55.20 ± 2.9[d] | 22.67 ± 0.51[de] | 3.23 ± 0.20[c] | 21.90 ± 1.04[cd] |
| TA | 56.03 ± 0.66[de] | 5.84 ± 0.47[bcd] | 22.93 ± 0.96[d] | 51.27 ± 3.61[d] | 22.00 ± 1.58[e] | 3.13 ± 0.15[c] | 20.06 ± 0.29[e] |
| TRA | 61.10 ± 1.22[b] | 6.30 ± 0.10[b] | 28.00 ± 0.17[b] | 67.83 ± 0.41[b] | 27.63 ± 0.55[b] | 3.90 ± 0.10[b] | 23.64 ± 0.21[b] |
| TK | 56.47 ± 0.37[cde] | 5.71 ± 0.32[cd] | 18.53 ± 0.25[f] | 43.27 ± 2.60[e] | 15.47 ± 1.51[g] | 2.37 ± 0.25[d] | 21.21 ± 0.12[d] |

Notes.

CK, Blank control; R, Inoculation with *R. intraradices*; A, inoculation with *A. calcoaceticus*; RA, inoculation with *R. intraradices* and *A. calcoaceticus*; TK, thiram spraying only; TR, thiram spraying and inoculation with *R. intraradices*; TA, thiram spraying and inoculation with *A. calcoaceticus*; TRA, thiram spraying and inoculation with *R. intraradices* and *A. calcoaceticus*.

Values are means ± standard error with three replicates.

Analysis of variance (ANOVA) and Duncan's test (honestly significant difference, HSD) were applied to evaluate significant differences ($P < 0.05$).

Different lowercase letters indicate significant differences from different treatments ($P < 0.05$).

## Effects of treatments on total bacterial colonies

The total number of colony forming units in the rhizosphere soil was greater in the group inoculated with *R. intraradices* and *A. calcoaceticus* than in the group inoculated with *R. intraradices*, the group inoculated with *A. calcoaceticus*, or the control group, and the number of bacterial colonies in the rhizosphere soil significantly increased by 46%, 83% and 133%, respectively ($P < 0.05$) (Table 1). The results revealed that the synergistic effect of *R. intraradices* and *A. calcoaceticus* increased the total number of bacterial colonies in the rhizosphere soil of the plants. Under the same inoculation treatment, the total number of bacterial colonies in the rhizosphere soil of the soybean plants in the thiram sprayed group was significantly lower than that in the non-sprayed thiram group ($P < 0.05$). The total number of bacterial colonies in the control treatment group was 94% greater than that in the thiram treatment group. The total number of bacterial colonies in the group inoculated with *R. intraradices* was 71% greater than that in the group sprayed with thiram and inoculated with *R. intraradices*. The total number of bacterial colonies in the group inoculated with *A. calcoaceticus* was 54% greater than that in the group sprayed with thiram and inoculated with *A. calcoaceticus*. The total number of bacterial colonies in the group inoculated with *R. intraradices* and *A. calcoaceticus* was 96% greater than that in the group sprayed with thiram and inoculated with *R. intraradices* and *A. calcoaceticus*.

## Effects of treatments on thiram residues

Compared with other treatment groups, the *R. intraradices* and *A. calcoaceticus* inoculation treatment that received thiram had the lowest residual amount of thiram in soybean grains and the rhizosphere soil (Table 3). For the soybean grain samples, in the group that was sprayed with thiram and inoculated with *R. intraradices* and *A. calcoaceticus*, relative to the group sprayed with thiram and inoculated with *R. intraradices*, the group sprayed with thiram and inoculated with *A. calcoaceticus*, or the group sprayed with only thiram, thiram residues were reduced by 44%, 57%, and 73%, respectively. For the

**Table 3  Effects of treatments on thiram residues in soybean grains and rhizosphere soil.**

| Treatments | Thiram residue in soybean grains (μg/mL) | Thiram residue in rhizosphere soil (μg/mL) |
|---|---|---|
| CK | 0.00 ± 0.00[e] | 0.00 ± 0.00[e] |
| TK | 7.38 ± 0.02[a] | 7.27 ± 0.02[a] |
| TR | 3.56 ± 0.05[c] | 4.46 ± 0.03[b] |
| TA | 4.59 ± 0.01[b] | 3.82 ± 0.03[c] |
| TRA | 1.99 ± 0.04[d] | 2.26 ± 0.02[d] |

**Notes.**

CK, Blank control; TK, thiram spraying only; TR, thiram spraying and inoculation with *R. intraradices*; TA, thiram spraying and inoculation with *A. calcoaceticus*; TRA, thiram spraying and inoculation with *R. intraradices* and *A. calcoaceticus*.

Values are means ± standard error with three replicates.

Analysis of variance (ANOVA) and Duncan's test (honestly significant difference, HSD) were applied to evaluate significant differences ($P < 0.05$).

Different lowercase letters indicate significant differences from different treatments ($P < 0.05$).

rhizosphere soil samples, in the group that was sprayed with thiram and inoculated with *R. intraradices* and *A. calcoaceticus*, relative to the group sprayed with thiram and inoculated with *R. intraradices*, the group sprayed with thiram and inoculated with *A. calcoaceticus*, or the group sprayed with only thiram, the thiram residues decreased by 49%, 41%, and 69%, respectively.

## DISCUSSION

Spore density reflects the ability of AMF to reproduce in the soil (*Birhane et al., 2021*). As shown in Table 1, inoculation with *R. intraradices* and *A. calcoaceticus* and spraying with thiram had significant effects on AMF spore density ($P < 0.05$). The AMF spore density significantly increased in the group inoculated with *R. intraradices* and *A. calcoaceticus* ($P < 0.05$), indicating that *R. intraradices* and *A. calcoaceticus* synergistically participated in soil material cycling, improved soil fertility, promoted plant root growth and development, and increased AMF spore density. Microbial inoculation counters thiram-induced damage by restoring bacterial diversity (thiram reduces bacterial colonies in soybean rhizosphere soil) through introducing beneficial strains, which compete with pathogens, degrade thiram residues *via* enzymatic detoxification, and rebuild soil structure *via* exopolysaccharides. These inoculants also enhance plant-microbe symbiosis, offsetting thiram's disruption of the soil microenvironment (*Sherif, Elhussein & Osman, 2011*). The synergistic effect of AMF and PSB can improve phosphorus absorption, AMF spore density, AMF infection rates, and plant growth (*Artursson, Finlay & Jansson, 2006*; *Nacoon et al., 2021*; *Toro, Azcón & Barea, 1997*). Inoculation with *R. intraradices* and *A. calcoaceticus* can lead to a more stable fungal-bacterial system, increase nutrient absorption by plants, inhibit soil pathogens, and maintain a low incidence of root rot. AMF can induce the synthesis of endogenous signaling substances in plants, including phytohormones such as abscisic acid (ABA), ethylene (ET), jasmonic acid (JA), and salicylic acid (SA), which activate plant defense mechanisms to enhance disease resistance (*Bortolot et al., 2024*). *Wang et al. (2022)* showed that AMF colonization upregulated JA synthesis gene expression in plants, leading to enhanced resistance against *F. oxysporum* with increased polyphenol

oxidase and phenylalanine ammonia lyase activities. Furthermore, AMF interact with plant growth-promoting rhizobacteria to increase phenolic compound secretion and promote lignification of cell walls, thereby inhibiting *Fusarium oxysporum* infection in plant roots (*Wang et al., 2018*). In the process of microbial induced disease resistance, the JA/ET pathway and SA pathway serve as the primary signaling pathways involved in plant defense against pathogens (*Jamil et al., 2022*). In this study, the number of soybean rhizosphere nodules in the group inoculated with *R. intraradices* and *A. calcoaceticus* was the greatest, indicating that a synergistic effect promoted the absorption of plant nutrients and thus increased the number of soybean rhizosphere nodules. Inoculation with *R. intraradices* or *A. calcoaceticus* can increase soybean biomass, indicating that a synergistic effect creates a plant-fungal-bacterial growth-promoting system that is more conducive to the absorption and utilization of soil nutrients by plants, thus promoting the growth of crops. In addition, spraying thiram increased soybean biomass significantly ($P < 0.05$), relative to the control group, indicating that thiram restrained pathogen growth in the soil, reduced the incidence of disease, and promoted crop growth. However, thiram also inhibited or killed beneficial soil microorganisms (*e.g.*, rhizobia, AMF), which resulted in significantly lower biomass in the thiram treatment than in the mixed inoculation treatment group ($P < 0.05$).

The PSB strain *Acinetobacter pittii* significantly increased the abundance of genes related to bacterial inorganic and organic phosphorus cycling in the soil (*He & Wan, 2021*). Since PSB depend on the carbon released by AMF, they secrete phosphatase and phytase to hydrolyse organophosphorus, which provides Pi for hyphal growth in AMF (*Etesami, Jeong & Glick, 2021*). The interaction between AMF and PSB was shown to increase the phosphorus uptake of bicolor sorghum and the total number of bacterial colonies in rhizosphere soil (*Calvo, Nelson & Kloepper, 2014*). In this study, the greatest number of bacterial colonies were found in the rhizosphere soil of soybean plants that were inoculated with *R. intraradices* and *A. calcoaceticus*, indicating that the synergistic effect of the two species increased the number of bacteria in rhizosphere soil, improved the microbial community structure, and improved the utilization of soil available phosphorus by plants, which increased soybean yield. However, spraying thiram reduced the total number of bacterial colonies in the rhizosphere soil of soybean plants, indicating that thiram inhibited or killed beneficial microorganisms when it entered the soil and that its long duration in the soil had adverse effects on the soil environment, which resulted in a significant decrease in the total number of bacterial colonies in the rhizosphere soil of soybean plants. The accumulation of pesticides in soil affects the activities of hydrolases, oxidoreductases, dehydrogenases, and phosphatases, and subsequently affects soil fertility (*Zhang & Yang, 2021*). AMF can degrade pesticide residues in soil and crops, and have a positive impact on food safety and ecological restoration (*Zhang et al., 2019*). After atrazine-contaminated soil was inoculated with AMF, the atrazine degradation rate was 91% (*Song et al., 2010*). In addition, *Kumar, Lakshmi & Khanna (2008)* reported that the PSB strain *Pseudomonas* sp. degraded residual endosulfan in the soil. After thiram spraying, we found that different degrees of residue occurred in the soybean grains and rhizosphere soil in the different treatment. Thiram residues in the soybean grains and rhizosphere soil sprayed with thiram and inoculated with *R. intraradices* and *A. calcoaceticus* decreased by 73% and 69%,

respectively, compared to the treatment that only sprayed thiram. These results indicated that *R. intraradices* and *A. calcoaceticus* degraded thiram. The goals of this study were to improve soybean yield and reduce pesticide residues, provide a basis for pesticide pollution remediation in agricultural ecosystems, and promote the sustainable development of agriculture.

## CONCLUSION

This study demonstrated that microbial inoculation enhanced soybean growth parameters by synergistically improving arbuscular mycorrhizal symbiosis, increasing total bacterial colonies, and accelerating pesticide degradation. The combined application of *R. intraradices* and *A. calcoaceticus* effectively inhibited root rot pathogens while optimizing plant-microbe interactions for nutrient acquisition. This dual-functional approach solved both soil-borne diseases and pesticide residues, providing a sustainable strategy to enhance soil fertility and crop yield. For agricultural integration, field validation should prioritize regionally adapted microorganisms and their compatibility with precision agriculture technologies. Future studies should focus on deciphering microbial community dynamics under different soil types and climatic variability, evaluating scalability in diverse agroecosystems, and conducting systematic comparisons between bioaugmentation and conventional chemical-dependent practices to advance ecological intensification frameworks.

## ACKNOWLEDGEMENTS

The authors are grateful to Engineering Research Center of Agricultural Microbiology Technology, Ministry of Education & Heilongjiang Provincial Key Laboratory of Plant Genetic Engineering and Biological Fermentation Engineering for Cold Region & Key Laboratory of Microbiology, College of Heilongjiang Province & School of Life Sciences, Heilongjiang University for providing the equipment for this study.

### Funding

This work was supported by a grant from the Natural Science Foundation of Heilongjiang Province (LH2023C087). The funders had no role in study design, data collection and analysis, decision to publish, or preparation of the manuscript.

### Grant Disclosures

The following grant information was disclosed by the authors:
The Natural Science Foundation of Heilongjiang Province: LH2023C087.

### Competing Interests

The authors declare there are no competing interests.

## Author Contributions

- Weiguang Jie conceived and designed the experiments, analyzed the data, authored or reviewed drafts of the article, and approved the final draft.
- Yiwen Tan performed the experiments, analyzed the data, prepared figures and/or tables, software, and approved the final draft.
- Houze Lin conceived and designed the experiments, prepared figures and/or tables, and approved the final draft.
- Min Zhang performed the experiments, prepared figures and/or tables, and approved the final draft.
- Lianbao Kan analyzed the data, authored or reviewed drafts of the article, software, and approved the final draft.

## Data Availability

The raw measurements are available in the Supplementary File.

## Supplemental Information

Supplemental information for this article can be found online at http://dx.doi.org/10.7717/peerj.19701#supplemental-information.

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
