# Peer review of "Effectiveness of *Rhizophagus intraradices* and *Acinetobacter calcoaceticus* on soybean growth and thiram residues in soybean grains and rhizosphere soil"

_PeerJ, doi:10.7717/peerj.19701_

## Round 0.1 · original submission · Major Revisions

Please answer all reviewers´comments.

·

Basic reporting

Overall, the writing style and use of English is correct with some suggestion noted below. A couple of issues should be dealt with however. The authors state they use an aqueous solution of pesticide but that substance is poorly soluble in water. I assume they used a suspension but this has to be stated. Significant figures refer to the values of their results to .01 decimals. Going to this significance is all right overall, but is it actually that value. Their methodologies were all selected and carried out properly. The authors apply pesticide directly to their pots and not coat the seeds as is usually done commercially. The reason for this is needed. Using 50 g of inoculum must also be further defined. Overall the research is valid and needs to be published once corrections are made.

Experimental design

The investigators studied the synergistic effect of how two bacterial species increased the total number of bacteria in rhizosphere soil, improved the overall microbial community structure, and improved the utilization of soil-available phosphorus by soybean plants increasing soybean yield. Spraying with pesticide reduced the total number of bacterial colonies in the rhizosphere soil of the plants, indicating that thiram inhibited or killed beneficial microorganisms when it entered the soil and that its long duration in the soil had adverse effects on the soil environment, which resulted in a significant decrease in the total number of bacterial colonies in the rhizosphere soil of soybean plants. The accumulation of pesticides in soil affected the activities of hydrolases, oxidoreductases, dehydrogenases, and phosphatases, and subsequently affected soil fertility.

Validity of the findings

The investigators used proper statistical methods in reaching their conclusions. The work is well referenced but there are a few issues which need further clarification.

Additional comments

L. 27. The whole thesis here is about soybeans so maybe “soybean” should be included here.

L. 34. Yield should be plural

L. 92. was not were

L. 101-102. The word between is ambiguous here. It means between two values but only one value is given here.

L. 107-108. Chitin is not part of microbial life but rather is what gives stems structure in plants. This sentence relates chitin to bacteria.

L. 119. Soybean seeds? Not plants….since 3 seedlings were left AFTER germination.

L. 124-124. 50g in oculum seems pretty high. Wet weight? Dry weight seems impossible. I’ve grown lots of bacteria in large bioreactors and this amount would be really difficult to generate easily. Do you mean 50 g bacterial suspension? About 50 mL?

L. 130. Impossible because Thiram is barely soluble in water as per this reference: In water, 30 mg/L at 25 °C. Yalkowsky, S.H., He, Yan., Handbook of Aqueous Solubility Data: An Extensive Compilation of Aqueous Solubility Data for Organic Compounds Extracted from the AQUASOL dATAbASE. CRC Press LLC, Boca Raton, FL. 2003., p. 299. I suspect you used a suspension in water.
https://pubchem.ncbi.nlm.nih.gov/compound/Thiram#section=ICSC-Number

L. 156. Dry et or wet et?

L. 199. Are the significant figures to .01% appropriate?

As an afterthought here, the chemical agent was added directly to the soil, but in practice, isn’t it normal to treat the seeds with thiram before placing them in the ground? So, if this experiment really meant to model the real situation as initially stated?

L.218-220. Need rewording.
The AMF infection rate in the group inoculated with R. intraradices andA. calcoaceticus was 12.96% (significantt figures again, to two places?) higher than that in the group subject to thiram spraying and inoculated with R. intraradices and A. calcoaceticus. This may be because thiram causes toxicity to the rhizosphere microorganisms of soybean roots, resulted resulting in the decrease of the AMF infection rate.

L. 224. Not “treatment”. treated plants

L. 225”. and the

L. 270. We usually refer to bacterial colonies as “colony forming units” or “cfu’s”

L. 278. sprayed

L. 361. increased the number of genes? This really does not make sense as worded. Genes are in organism DNA, not in the soil.

L. 367. Ma et al[42] is the normal way to reference when giving the author’s name.

L. 407. possible solution

L. 606. Different letters indicate significant differences from different treatments (p < 0.05). What does this mean? Here and in the following tables. In tables, terms have to be exact and unambiguous.

Reviewer 2 ·

Basic reporting

The manuscript presents " Effectiveness of Rhizophagus intraradices and Acinetobacter calcoaceticus on Soybean Growth and Thiram Residues in Soybean Grains and Rhizosphere
Soil" an investigation into the application of Rhizophagus intraradices and Acinetobacter calcoaceticus singly or in combination as well as with Thiram as bio-inoculants in soybean cultivation. Authors looked into the plant growth, microbial impact and thiam residues. While the study addresses an important topic, I found several shortcomings that need to be addressed to improve the paper.

In case of the writing quality, the overall writing is weak, with frequent irrelevant and misleading references. The introduction contains multiple incorrect citations and redundant information. The results section is overly descriptive and often mixed with discussion points. To me, the data could be presented more concisely under three headings: Effect of treatment on plant growth parameters, Effect of treatment on microbial population, Effect of treatment on thiram residue degradation. Similarly, the discussion is excessively long and includes unnecessary details. Only key findings with appropriate references should be highlighted.

Experimental design

While the study addresses an important topic, I found several shortcomings that need to be addressed to improve the paper.
For example, the identity of R. intraradices and A. calcoaceticus was not confirmed using molecular techniques, which are essential for accurate microbial identification. The absence of molecular confirmation limits the reliability of the microbial strains used.
The study was conducted solely in pot experiments without validation in field conditions. Field conditions can vary significantly in terms of soil composition, microbial diversity, environmental stress, and other abiotic factors. Many bio-inoculants that perform well in vitro or pot trials often fail in natural field conditions. I recommend conducting several repeated field trials, even on small plots in experimental fields, to validate the efficacy of the bio-inoculants under realistic conditions.
The study relied on colony count and isolation-based techniques, which provide only a minute fraction of the whole microbial community. A comprehensive microbiome analysis using next-generation sequencing (NGS) or other molecular methods would offer a more robust understanding of plant-microbe interactions.
The manuscript did not investigate whether microbial inoculation influenced plant physiology at the molecular or gene expression level. Including gene expression analysis of key plant defense or growth-related genes would strengthen the mechanistic understanding of the plant-microbe interaction.

Validity of the findings

no comment

Additional comments

While checking at a glance, I found some issues but there could be more. For example
• L36: Revise to a more precise expression such as "a range of pathogenic fungi are involved."
• L39-40: Include additional pathogenic genera such as Phytophthora, Pythium, Rhizoctonia, and Sclerotium.
• L42: Thiram is not considered a broad-spectrum antibacterial agent in crops. If claimed, provide appropriate references.
• L44: Reference 6 does not specifically discuss thiram. Reference 7 provides general toxicity information but not specific to thiram. Ensure correct citation.
• L54: The cited reference does not estimate that 80% of plants have AMF. Replace or omit.
• L59: The citation is incorrect—Nell et al., 2010, not Berdeni et al., and not related to apple.
• L79: This citation is Berdeni et al., 2018 and pertains to apple, not soybean root rot.
• L62: Add "For example" before listing examples.
• L67: If the cited paper does not discuss the phytase gene, replace or omit.
• L80: Mention which studies support the claim.
• L91: Replace "sensitive" with "susceptible."
• L92: Replace "were to" with "was."
• L94: Cite the protocol for species identification.
• L94: Explain how species identity was confirmed.
• L98: Provide detailed isolation and characterization methods.
• L103-109: Delete this section as it is unnecessary.
• L111: Specify pot size, shape, and volume.
• L118: Clarify whether 10 pots per treatment with 3 plants per pot were used.
• L190: Report standard error instead of standard deviation.
• L193-303: Rewrite this section more concisely.
• L305-399: Streamline this section with only key findings.
• L309-310: Delete this sentence as it is redundant.

Reviewer 3 ·

Basic reporting

The main focus of the article is to investigate the effects of Rhizophagus intraradices and Acinetobacter calcoaceticus inoculations on soybean growth, root rot incidence, and the reduction of thiram residues. The study addresses a highly relevant and current research topic by aiming to support agricultural sustainability, provide biological control against plant diseases, and reduce chemical pesticide residues. The research hypothesis suggests that the combination of R. intraradices and A. calcoaceticus will promote soybean growth, reduce root rot disease, and significantly decrease thiram residues in both soybean grains and rhizosphere soil. This hypothesis is scientifically sound and practical, aiming to highlight the potential benefits of biological fertilizers and microbial inoculations in agricultural production.

Experimental design

The experimental design is well-structured and appropriate for testing the research hypothesis. The study effectively compares the effects of Rhizophagus intraradices and Acinetobacter calcoaceticus inoculations, both individually and in combination, along with thiram application. The inclusion of multiple treatment groups enhances the comprehensiveness of the study. The controlled pot experiment minimizes environmental variability, and the chosen parameters (AMF spore density, infection rate, root rot incidence, nodule number, plant biomass, bacterial colonies, and thiram residues) provide a thorough evaluation of plant growth and soil microbial dynamics.

However, the study's limitation lies in its controlled conditions, which may not fully represent field applications. Future research should explore different soil types and climatic conditions to assess real-world applicability. Clearly specifying the statistical methods used (e.g., ANOVA, Tukey HSD test) in each table would further strengthen the study’s validity. Overall, the experimental design is robust and well-planned, with potential for future refinement through field trials.

Validity of the findings

The validity of the findings is supported by a well-structured experimental design, appropriate statistical analyses, and consistent results across multiple treatment groups. The study effectively demonstrates the impact of Rhizophagus intraradices and Acinetobacter calcoaceticus inoculations on soybean growth, root rot incidence, and thiram residue degradation. The use of control and treatment groups, along with replication, enhances the reliability of the results.

However, while the study provides strong evidence under controlled conditions, field trials in diverse environmental settings would further validate the applicability of the findings. Additionally, specifying the statistical tests used (e.g., ANOVA, Tukey HSD test) in all result presentations would strengthen the robustness of the conclusions. Overall, the findings are scientifically valid, but future research in real-world conditions is recommended to confirm their broader applicability.

Additional comments

Overall, the article appears to be suitable for publication in a scientific journal. The topic is current, the research hypothesis is scientific, and the experimental design is appropriate. However, minor improvements in the abstract, results, discussion, and conclusion sections would further enhance the scientific quality of the manuscript. Clearly stating the statistical analyses used, objectively presenting the study’s limitations, and including more detailed future research suggestions are recommended. Therefore, the article is deemed suitable for publication with minor revisions.

Annotated reviews are not available for download in order to protect the identity of reviewers who chose to remain anonymous.

·

Basic reporting

Nicely written, require minor revisions languages and length of the paper.

Experimental design

Appropriate

Validity of the findings

No Comment

Additional comments

No Comment

---

## Round 0.2 · Major Revisions

Please address the reviewer comments.

Reviewer 3 ·

Basic reporting

In the study, it was more effective with the arrangements made to the areas to be corrected.

Experimental design

The deficiencies mentioned in experimental design have been eliminated. The article has been more appropriate in this form.

Validity of the findings

uygundur.

·

Basic reporting

Kindly provide the base line data of the soil and Thiram residue for better comparison, as soil is collected from soybean field.

Experimental design

Kindly provide ANOVA of the experimental design.

Validity of the findings

Can not be sure as this is pot experiment.

Additional comments

Table 1 and Table 3 can be merged.

---

## Round 0.3 · accepted · Accept

Thanks for addressing all comments!